# The Implementation of Antimicrobial Consumption Surveillance and Stewardship in Human Healthcare in Post-Soviet States: A Systematic Review

**DOI:** 10.3390/antibiotics14080749

**Published:** 2025-07-25

**Authors:** Zhanar Kosherova, Dariga Zhazykhbayeva, Ainur Aimurziyeva, Dinagul Bayesheva, Yuliya Semenova

**Affiliations:** 1School of Medicine, Nazarbayev University, Astana 010000, Kazakhstan; zbeysenova@nu.edu.kz (Z.K.); yuliya.semenova@nu.edu.kz (Y.S.); 2School of Sciences and Humanities, Nazarbayev University, Astana 010000, Kazakhstan; ainur.aimurziyeva@nu.edu.kz; 3Department of Paediatric Infectious Diseases, NJSC “Astana Medical University”, Astana 010000, Kazakhstan; baesheva.d@amu.kz

**Keywords:** anti-bacterial agents, anti-infective agents, drug utilisation, antimicrobial stewardship, humans, public health, global health, Armenia, Azerbaijan, Belarus, Estonia, Georgia, Kazakhstan, Kyrgyzstan, Latvia, Lithuania, Moldova, Russia, Tajikistan, Turkmenistan, Ukraine, Uzbekistan

## Abstract

**Background/Objectives:** Antimicrobial consumption (AMC) surveillance and antimicrobial stewardship (AMS) constitute effective strategies to combat the increasing antimicrobial resistance rates worldwide. Post-Soviet countries (Armenia, Azerbaijan, Belarus, Estonia, Georgia, Kazakhstan, Kyrgyzstan, Latvia, Lithuania, Moldova, the Russian Federation, Tajikistan, Turkmenistan, Ukraine, and Uzbekistan) implemented various elements of AMC surveillance and AMS to different extents. The limited quantity and quality of data from post-Soviet countries make it difficult to assess health system performance; therefore, this region is a blind spot in global AMR monitoring. This systematic review assesses and characterises AMC surveillance and AMS implementation in post-Soviet countries. **Methods:** Evidence was compiled via a search in PubMed, Google Scholar, Embase, CyberLeninka, and Scopus. The eligibility criteria included AMC surveillance- and AMS-related papers in human health within defined regions and timelines. Some literature from the official websites of international and national health organisations was included in the search. **Results:** As a result of the searches, screening, and critical appraisal, three peer-reviewed publications and 31 documents were selected for analysis. Eleven out of fifteen countries with updated national action plans for combating antimicrobial resistance have defined AMC surveillance and AMS as strategic objectives. All 15 examined countries submitted antimicrobial consumption data to international networks and reported the existence of approved laws and regulations on antibiotic sales. However, disparities exist in the complexity of monitoring systems and AMS implementation between high-income and low-income countries in the region. **Conclusions:** This review provides key insights into the existing AMC surveillance and AMS implementation in former Soviet countries. Although the approach of this review lacks quantitative comparability, it provides a comprehensive qualitative framework for national-level AMC surveillance and AMS system assessment.

## 1. Introduction

Antimicrobial resistance (AMR) is a global health concern. Estimates project that by 2050 there could be approximately 1.91 million annual deaths attributable to AMR and 8.22 million AMR-associated deaths [1,2]. The bulk of antimicrobial resistance consists of anti-bacterial resistance. The reason for this is that bacteria are the most common causative agents of infections, and they have developed diverse mechanisms to evade the effects of antibiotics. Increasing rates of AMR are driven by the extensive use of antibiotics in both animal and human health sectors. Addressing this complex challenge requires the use of the One Health approach—a collaborative, multisectoral framework that coordinates antimicrobial resistance interventions across human, animal, and environmental health sectors. Although most of the antibiotic use burden is related to animal health, the use of antibiotics in human health should be addressed [3].

To address the global rise of AMR, antimicrobial consumption (AMC) surveillance and antimicrobial stewardship (AMS) have proven to be effective tools and approaches [4]. The purpose of AMC surveillance is to quantify and examine trends in antibiotic consumption in various healthcare environments. Moreover, consumption data assists in identifying gaps where AMS initiatives, along with legislative and policy measures, can be most successfully implemented [5]. Complementarily, AMS plays a crucial role in managing AMR by ensuring that antimicrobials are used appropriately and effectively. By promoting the right drug, dose, and duration, AMS helps preserve the effectiveness of antibiotics in the future [6,7].

Given the various extents of actions taken by countries worldwide, examining specific regional contexts becomes crucial, particularly in areas with unique healthcare system characteristics. Today, the post-Soviet region includes countries with varying income levels, a factor believed to significantly influence their capacity to implement AMC surveillance and AMS [7]. Moreover, income level primarily affects resource availability and infrastructure [8]. The limited quantity and quality of data from former USSR countries (Union of the Soviet Socialist Republics) make it difficult to assess health system performance; therefore, this region is a blind spot in global AMR monitoring [7,9]. Although these countries share a common history and legacy, their healthcare performance varies significantly due to differences in healthcare reforms and political and economic factors. Since implementation strongly depends on health system structure and mechanisms, it is crucial to establish a thorough understanding in order to implement country-tailored effective approaches [10]. Although a previous systematic review characterised AMR surveillance in the region [11], little is known about AMC surveillance and AMS establishment in post-Soviet Union countries. This systematic review aims to assess and characterise existing AMC surveillance and AMS implementation in post-Soviet countries.

## 2. Results

### 2.1. Publication Selection

Six hundred and eighty publications were found during the initial database search. A total of 73 duplicates were removed, leaving 607 articles for screening and critical appraisal. Title, abstract, and full-text screening narrowed the eligible publications to seven after excluding geographically irrelevant articles, AMR surveillance studies, physician surveys, population surveys, and clinical aspects related to antibiotics. These seven papers were assessed using the Joanna Briggs Institute (JBI) checklist, resulting in three publications being included in the review.

The grey literature search identified 38 documents from health organisations and websites. Of these, 10 were excluded due to the unavailability of the full text, orders not relevant to AMR surveillance systems, responsibilities of clinical pharmacologists, awareness surveys among physicians and the population, and policy briefs. As a consequence, 28 files were compiled from national and international health organisations.

To conclude, the selected 31 documents included 11 updated national action plans (NAPs) (36.7%), one report (3.3%) compiled in the Tracking Antimicrobial Resistance Country Self-Assessment Surveys (TrACSS) for 2023, 16 publicly available reports (50%) compiled and developed by international organisations, including the World Health Organisation (WHO), European Centre for Disease Prevention and Control (ECDC), Global Antimicrobial Resistance and Use Surveillance System (GLASS), and European Surveillance of Antimicrobial Consumption Network (ESAC-Net), and three (10%) publications from peer-reviewed journals. A list of the included studies is provided in Appendix A.

### 2.2. Antimicrobial Consumption Surveillance

Table 1 presents the results of national-level AMC surveillance, including (i) the prioritisation of AMC in NAP; (ii) established laws/regulations on antimicrobial prescription/sales in human health; (iii) the utilisation of AMC data to guide decision-making; (iv) reporting to international networks.

In terms of NAPs, 14 countries, excluding Estonia, reported having developed and approved NAPs in the TrACSS 2023 response. However, 4 out of the 14 NAPs were not identified despite a thorough search. Eleven countries (73%)—Armenia, Georgia, Kazakhstan, Kyrgyzstan, Latvia, Lithuania, Moldova, the Russian Federation, Tajikistan, Turkmenistan, and Ukraine—had publicly available and up-to-date NAPs. According to our analysis, all 11 countries identified AMC surveillance as a strategic objective (Appendix A).

Importantly, all countries have established laws and regulations governing the prescription and sale of antimicrobials. Moving on to the use of AMC data to guide decision-making, Georgia is the only country that reported not using antimicrobial consumption or using data to modify laws and regulations in the country.

According to Table 1, 14 (93.3%) countries—all except Turkmenistan—report antimicrobial consumption data to international networks such as the WHO AMC network, GLASS, and ESAC-Net. Estonia, Latvia, and Lithuania (20%) share consumption data with ESAC-Net and GLASS. The Russian Federation, Armenia, Belarus, Moldova, Ukraine, and Tajikistan (40%) share data with the WHO AMC Network and GLASS. Finally, Azerbaijan, Georgia, Kazakhstan, Kyrgyzstan, and Uzbekistan report consumption data only to the WHO AMC Network.

Figure 1 reveals significant disparities in AMC surveillance implementation across the fifteen reviewed countries. The analysis demonstrates that less than half of the examined countries (40%) remain at the foundational level, suggesting substantial room for improvement in AMC surveillance capabilities. Notably, only 3 of the 15 countries (20%) have progressed to comprehensive monitoring systems, underscoring that these nations can provide the most representative and consistently collected data. This uneven distribution highlights the need for targeted capacity-building initiatives to strengthen surveillance systems, particularly in countries with foundational-level implementation.

### 2.3. Antimicrobial Stewardship

Table 2 reveals critical gaps in AMS implementation across three key domains: strategic prioritisation, antimicrobial use optimisation, and AWaRe classification adoption. While nearly three-quarters of countries have prioritised AMS strategically, other elements vary significantly. Most concerning is that only four countries have achieved comprehensive antimicrobial use optimisation, with the majority remaining at partial implementation levels. Additionally, AWaRe classification adoption remains split, with slightly less than half of countries failing to integrate this WHO framework into their National Essential Medicines Lists. These findings highlight substantial implementation gaps despite the widespread strategic recognition of AMS importance.

### 2.4. Subgroup Analysis

The findings revealed various levels of AMC surveillance and AMS implementation across the examined region.

#### 2.4.1. Similarities Across Groups

All examined countries with NAPs defined both AMC surveillance and AMS as strategic objectives. Although the structure and content of the NAPs varied, we could differentiate between the strategic objectives and simple activities and events (Appendix A). As described in the results section, four NAPs were not retrieved even though their development and approval were reported. The possible reason for this is that these NAPs were used solely for internal purposes; however, the public accessibility of NAPs is an important indicator of transparency and accountability [1]. NAPs are crucial documents for addressing AMR and establishing related legislative frameworks, including policies and national guidelines on antibiotic use and prescription. However, the mere availability of NAPs does not guarantee that specific strategies against AMR will be effectively implemented or enforced [3].

All examined countries, except Turkmenistan, report consumption data to international networks according to region.

#### 2.4.2. Differences Across Groups

According to the results, the AWaRe classification is more integrated in upper-middle-income countries (UMICs) than in HICs and LMICs. Nevertheless, Latvia stated in its NAP [23] the necessity to align its National Essential Medicines List with WHO recommendations. Latvia lacks antibiotic consumption monitoring in hospitals, whereas it has lower than average rates of antibiotic consumption among European Union countries [23]. Lithuania has regulations for antimicrobial consumption monitoring [24].

According to Table 2, Kyrgyzstan reports on the implementation of AMS and suggests strategies to promote the appropriate use of antibiotics at the community and hospital levels. In terms of the establishment of AMC surveillance components, Kyrgyzstan, Uzbekistan, and Tajikistan report the same level, which allows for monitoring antibiotic consumption and sales.

When comparing AMC surveillance implementation, HICs mostly have developed levels and LMICs mostly report foundation levels. At the same time, UMICs report a mix of foundation, developing, and comprehensive levels of implementation.

## 3. Discussion

This research has a key strength as it is the first systematic review that synthesises published evidence on national AMC surveillance and AMS systems in post-Soviet countries. The findings reveal that income level is related to implementation capacity in post-Soviet countries. The HICs and UMICs in the region mostly have a higher implementation status than the LMICs. Several studies have confirmed this observation for HICs and LMICs worldwide [25,26,27]. For example, a study exploring global antibiotic consumption compared rates among countries with different income levels. The study showed that global antibiotic consumption in human health rose by 16.3%, with UMICs and LMICs making the highest contributions. Interestingly, HICs showed stable low consumption rates over the study period, even during the COVID-19 pandemic [25]. The decreasing antibiotic use trend in HICs might be explained by effective local initiatives and high commitment and support [28]. Another study revealed disparities in formal AMS programme implementation between LMIC and HIC healthcare settings. According to this study, only 42% of LMIC facilities reported established programmes compared to 76% of HIC facilities [26]. Thus, in general, LMICs have a lower implementation capacity for AMC surveillance and AMS, and the next section explores the potential reasons for this in this regional context.

Low levels of political commitment represent significant barriers to AMC surveillance and AMS adoption in LMICs. This lack of commitment is primarily attributed to insufficient funding and limited expertise in organising AMR initiatives [29]. Another challenge overlapping with the lack of finance and knowledge is fragmented healthcare systems and infrastructure. In addition, most of the former USSR states inherited a centralised decision-making approach and rigid public health structures. The existence of separate vertical public health structures has led to inefficiencies and the duplication of services, which adds to the fragmentation of the system [30]. These issues lead to problems at the institutional level, such as limited personnel, a lack of equipment, a lack of capacity in microbiology laboratories, and a lack of acknowledgement of AMS’s necessity by senior physicians [26,27,31]. The scarcity of good-quality representative data makes it difficult to provide the necessary information for prioritising AMS programmes [32]. These overlapping and consequent barriers result in a reactive type of implementation. This type is consultation-based rather than deploying proactive antimicrobial oversight. This type of approach may result in missed opportunities for early intervention and the optimisation of antimicrobial therapy, particularly in cases where prescribing patterns could be improved before resistance or adverse outcomes develop [33]. Therefore, the implementation of effective measures to combat AMR in LMICs requires changes in fundamental structural issues, such as infrastructure, financial support, high-level commitment, and regulatory commitment [3,34].

Regarding regulatory frameworks, all countries reported having established laws and regulations on the prescription and sale of antimicrobials. Despite this, over-the-counter sale is still an issue in most countries, including Azerbaijan, Georgia, Kyrgyzstan, Tajikistan, the Russian Federation, and Kazakhstan [1,13,14,15,16,35]. For example, in Azerbaijan, pharmacies belong to private organisations, not to the Ministry of Healthcare, and there is wide access to antibiotics via the internet [13]. Similarly, Lithuania describes in its NAP [24] that there is limited state regulation of the pharmaceutical market through the compilation of lists of first-choice and reserve antimicrobial drugs. Moreover, almost all countries (except Georgia) reported using data on antibiotic consumption to guide decision-making and modify policies.

One unexpected finding is that HICs, such as Lithuania and Latvia, show similar levels of antimicrobial use optimisation to LMIC countries, such as Kyrgyzstan, Tajikistan, and Uzbekistan. This suggests that other factors, such as political prioritisation, the structures of healthcare systems, and historical approaches to antibiotic use, may also play equally important roles in determining success [36]. This also suggests that resource availability alone does not guarantee success. This pattern suggests that even HICs with substantial resources may struggle to maintain policy momentum and sustain the achieved indicators. Unlike other health initiatives, in which success builds momentum, AMR creates a “despair trap.” When countries observe increasing AMR rates despite their efforts, they may lose the political will to continue investing in resources. Struggles specific to HICs include competing priorities in the health sector. This means that AMR is competing with cancer, heart disease, and other health issues for political attention. The second is complacency, where initial improvements may lead to a false sense of confidence that the problem is “solved”. The third reason is complex healthcare systems, meaning that it is harder to coordinate actions across multiple private and public providers [37]. Importantly, poorer countries often maintain their efforts because AMR is seen as an immediate and visible crisis, as people are dying of untreatable infections. Moreover, less complex healthcare systems allow for more centralised responses [37]. Due to the retention of the elements of the Semashko model in the former USSR states, the overall structure and governance of healthcare systems in most of them are characterised by a strong vertical hierarchy, contributing to a centralised response. The Semashko model is a centralised, state-funded healthcare system that provides universal free medical services through a hierarchically organised network of public institutions [7]. Even countries with resources and expertise can lose momentum against AMR when faced with worsening trends, suggesting that sustaining long-term political commitment to AMR control requires different strategies than other public health challenges [37].

In conclusion, among developing economies, access to healthcare and pharmaceuticals has improved along with economic progress, but with the caveat of high rates of antibiotic misuse. Countries that implement strong, coordinated policies to combat antimicrobial resistance, such as surveillance programmes, public education, and antibiotic use regulations, tend to better control antibiotic overuse and resistance over time. Moreover, meaningful improvements require high levels of sustained effort, and half-measures are not enough. Thus, a country-specific, priority-tailored approach based on data is recommended [29,37,38,39,40].

### Limitations of Research and Future Implications

Although this systematic review provides valuable insights into AMC surveillance and AMS in post-Soviet countries, several limitations must be acknowledged. One of the limitations of the study is the heavy reliance on TrACSS, which might introduce potential response bias. To mitigate this limitation, the country responses were cross-validated with other publicly available resources, mainly Joint External Evaluations by the WHO.

Although antibiotic consumption mostly occurs in agriculture and has implications for human health, this manuscript focuses specifically on human health. Further research may be conducted to explore the integration of the One Health approach in combatting the issue of AMR.

Despite its limitations, this systematic review has strengths. To the best of our knowledge, this research represents the first step towards a deeper understanding of the systemic and strategic elements established by former USSR countries in the field of antimicrobial consumption surveillance and stewardship.

The potential implications of this review include future studies that could track changes in AMC surveillance and AMS systems over time in these countries. Another potential implication is that a more in-depth analysis and understanding of healthcare systems and an assessment of healthcare performance in AMC surveillance and AMS may be conducted. To achieve this, either the developed qualitative framework can be refined and validated, or the quantitative framework can be developed with standardised metrics for AMC surveillance and AMS systems comparison.

## 4. Materials and Methods

### 4.1. Protocol Registration

This systematic review was registered in the International Prospective Register of Systematic Reviews (PROSPERO) under the protocol CRD42024626245.

### 4.2. Eligibility Criteria

Publications from peer-reviewed journals and grey literature, including evidence from national and international organisations, were selected for this systematic review according to inclusion and exclusion criteria. The eligibility criteria within the PICO (Population, Intervention, Comparison, and Outcomes) framework are provided in Appendix A. Inclusion criteria: (i) Papers on antimicrobial consumption and antimicrobial stewardship in Armenia, Azerbaijan, Belarus, Estonia, Georgia, Kazakhstan, Kyrgyzstan, Latvia, Lithuania, Moldova, Russia, Tajikistan, Turkmenistan, Ukraine, and Uzbekistan (former USSR countries); (ii) NAPs on combating antimicrobial resistance in the examined countries; (iii) papers published from 1 May 2015 to 31 December 2024; (iv) studies describing regions where at least one of the countries listed above was included; (v) publications in the English and Russian languages; (vi) country assessments by international organisations or country self-assessments.

Exclusion criteria: (i) Veterinary or agricultural studies; (ii) laboratory and microbiology research papers; (iii) clinical interventions using antibiotic consumption and antibiotic stewardship; (iv) antiviral, antimycotic, and antifungal consumption; (v) conference posters and abstracts; (vi) MD (Doctor of Medicine), MPH (Master of Public Health), and PhD (Doctor of Philosophy) dissertations; (vii) geographically irrelevant studies; (viii) publications in local languages other than English and Russian; (iv) papers not relevant to the defined time frame (from 1 May 2015, to 31 December 2024).

### 4.3. Search Strategies

The research articles were searched for in PubMed, Google Scholar, CyberLeninka, and Scopus. The search strategy for PubMed was developed using subject headings (MESH) terms and Boolean connectors. The search strategy for Google Scholar was conducted using the following keywords: “antimicrobial consumption”, “antimicrobial stewardship”, “National Action Plan”, “AMR strategy”, “post-Soviet countries”, “USSR”, “Russian Federation”, “Russia”, “Kazakhstan”, “Uzbekistan”, “Kyrgyzstan”, “Tajikistan”, “Latvia”, “Lithuania”, “Estonia”, “Georgia”, “Armenia”, “Ukraine”, “Moldova”, “Belarus”, “Azerbaijan”, and “Turkmenistan”. For the Google Scholar search, the first ten pages of the initial search results were screened. Another search used the same keywords translated into Russian to expand on the scholarly papers in CyberLeninka, and the first 10 pages of the results were screened. The paper publication timeline started in May 2015 when the WHO launched the Global Action Plan on AMR and engaged country members in initiating strategic measures to combat AMR [41].

During the search for scholarly articles, the reviewers encountered a lack of required information about the AMC surveillance and AMS architectures, as well as up-to-date NAPs across the examined countries. To obtain more information about the structures of AMC surveillance and AMS in each country, evidence was searched for on the official websites of international and national healthcare organisations and agencies, including the WHO, ECDC, Ministries of Healthcare, and National Centres of Healthcare.

### 4.4. Study Selection

Findings from the search strategies in the databases and grey literature sources were exported to a reference manager account in Zotero (version 6.0.36), with subsequent duplicate removal. An initial list of articles was prepared. An Excel spreadsheet was generated for the titles, abstracts, and full-text screening. In the next step, two reviewers independently selected articles on the matter of alignment with the research questions and inclusion and exclusion criteria. Any concerns regarding the eligibility of the academic papers or grey literature sources during the screening were addressed with the academic supervisor. The Preferred Reporting Items for Systematic Reviews and Meta-Analysis (PRISMA) 2020 checklist (see Figure 2) was carefully followed during the selection stage. The PRISMA 2020 checklist has been uploaded and included in Appendix A.

### 4.5. Quality Assessment

Seven peer-reviewed articles and 38 files from international and national healthcare organisations were selected for critical appraisal using the previously mentioned Excel format.

The peer-reviewed articles were independently evaluated by two researchers using the Joanna Briggs Institute (JBI) appraisal checklist for qualitative studies [42]. Before the assessment, reviewers discussed the questions and determined the priority criteria for inclusion. For this checklist of ten questions, the reviewers were provided with the options “yes”, “no”, “partly”, and “not available”. Articles with five or more “yes” answers were considered for further analysis.

The AACODS (authority, accuracy, coverage, objectivity, date, and significance) checklist was applied by two reviewers independently to evaluate the grey literature sources [43]. For this checklist of six criteria, the reviewers were provided with the options “yes”, “no”, “party”, “unclear”, and “not available”. Priority criteria for inclusion, such as authority, date, and significance, were discussed and determined before the assessment process.

Any disagreements between the two reviewers were resolved in consultation with the supervisor.

All examined countries are official WHO members and have demonstrated their dedication to achieving the health goals set by the WHO. To track how much progress they have made towards implementing AMC surveillance and AMS, we used the elements mentioned in the WHO guidelines [4,5,40,44].

First, all NAPs were translated into English using an online translator (except for Tajikistan and Turkmenistan, which were available in English) for the analysis. The NAPs were analysed by two reviewers to assess the prioritisation of AMC monitoring and AMS. The reviewers agreed to classify the prioritisation of AMC surveillance and AMS as no prioritisation, activity, or strategic objective (Appendix A). Out of the 11 identified NAPs, 10 were up to date, and their expiration dates ranged from 2025 to 2030, while the NAP of Tajikistan expired in 2022.

Next, monitoring systems for antibiotic consumption, as shown in Figure 1, were categorised as follows: (1) foundational level; (2) developed level, and (3) comprehensive level. This categorisation was based on the countries’ responses to the TrACSS. The foundational level included responses from those with no system or national plan designed to support antimicrobial consumption monitoring. The developed level included responses from countries with a system in place allowing the monitoring of antibiotic consumption and sales, and a system intended to track antimicrobial sales with additional monitoring of antibiotic use at regional levels. Lastly, the comprehensive level included responses of systems designed to monitor a representative sample of healthcare facilities regarding rational antibiotic use and prescription practices and with representative data regularly collected and reported on antibiotic sales and consumption and antibiotic prescription and use in both the private and public sectors [12]. For Table 1, elements including laws and regulations on antibiotic sales, as well as the use of antimicrobial consumption data to guide decision-making and/or modify policies, were retrieved with the corresponding answers recorded as “yes” or “no.” The status of reporting data to international networks across the country was extracted from the corresponding reports of the WHO AMC Network, GLASS, and ESAC-Net [17,18,19,20].

Table 2 presents the elements for antimicrobial stewardship implementation at the national level. The antimicrobial prescription and use practices were retrieved from TrACSS responses and Joint External Evaluations and categorised as follows: (1) no implementation, (2) development and partial implementation, (3) comprehensive and sustained implementation. No implementation level referred to no or weak national policies for appropriate antimicrobial use. Next, development and partial implementation referred to the following: “National policies promoting appropriate antimicrobial use/antimicrobial stewardship activities developed for the community and health care settings” and/or “National guidelines for the appropriate use of antimicrobials are available, and antimicrobial stewardship programmes are being implemented in some healthcare facilities [12].” Lastly, the comprehensive and sustained level referred to the following evaluations: “National guidelines for appropriate use of antimicrobials are available and antimicrobial stewardship programs are being implemented in most health care facilities nationwide.” “Monitoring and surveillance results are used to inform action and to update treatment guidelines and essential medicines lists”. “National guidelines on optimising antibiotic use are implemented for all major syndromes and data on use is systematically fed back to prescribers” [12]. The integration of the AWaRe classification into the National Essential Medicines List was retrieved from TrACSS responses, Joint External Evaluations, and NAPs and categorised as either “yes” or “no”.

All findings were compared and discussed according to the World Bank income classification of the post-Soviet countries. In the course of data synthesis, the results table was populated exclusively with data and evidence drawn from the most recently dated documents, thereby ensuring that the analysis reflected the most current and relevant information available.

## 5. Conclusions

The systematic review reveals critical disparities in AMC surveillance and AMS implementation across post-Soviet countries, with HICs demonstrating comprehensive systems while lower-middle-income countries remain at lower levels. Urgent regulatory framework strengthening is essential, as over-the-counter antibiotic access persists in multiple examined countries, undermining existing policies. Public health authorities must prioritise enforcement mechanisms, digital health technologies for surveillance enhancement, and capacity building through coordinated investment to address gaps across the region.

Future research should focus on evaluating context-specific factors beyond income level that influence success, including political prioritisation, healthcare system structure, and historical approaches to antibiotic use. Innovative intervention models should include tiered surveillance systems with flexible implementation levels, regional collaboration networks for peer-to-peer learning and resource sharing, and technology-enabled solutions such as interoperable surveillance systems and mobile health platforms for extending reach.

The review demonstrates that income level alone does not guarantee implementation success, and sustained political commitment and long-term strategic planning are critical success factors. Blended financing approaches combining domestic and international resources should be developed alongside economic evaluation frameworks to demonstrate cost-effectiveness and support transition from external to domestic funding. The identified “despair trap” phenomenon—where increasing AMR rates despite efforts lead to decreased political will—must be countered through modular training programmes, collaboration initiatives, and regular policy review cycles that maintain momentum for this critical public health challenge.

## Figures and Tables

**Figure 1 antibiotics-14-00749-f001:**
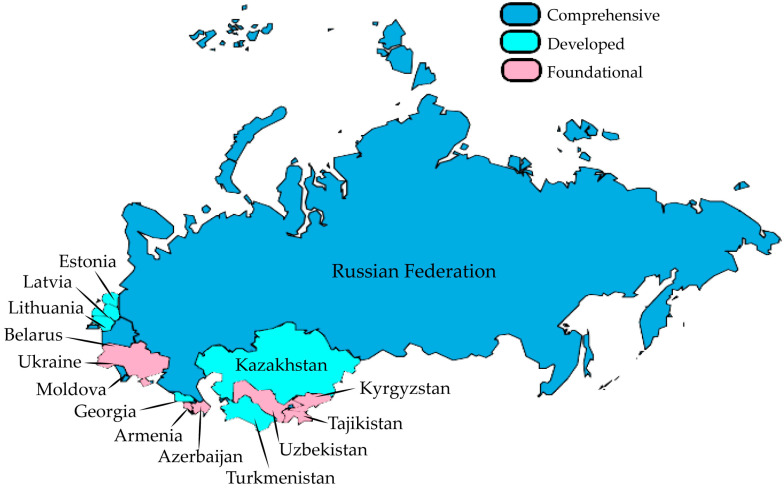
Monitoring systems for consumption and rational use of antimicrobials for human health in post-Soviet countries.

**Figure 2 antibiotics-14-00749-f002:**
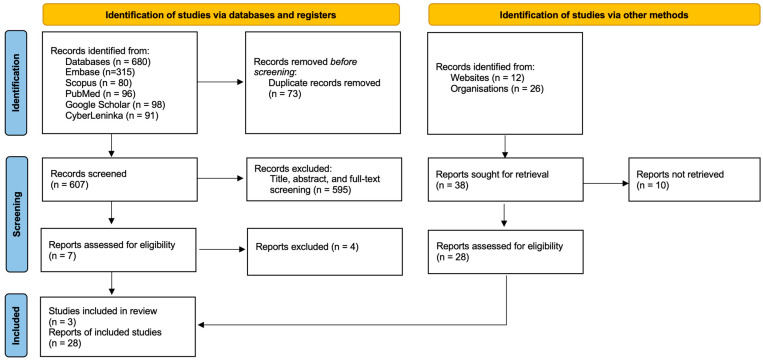
PRISMA 2020 flow diagram 4.6. Data synthesis.

**Table 1 antibiotics-14-00749-t001:** Antimicrobial consumption surveillance at national level in post-Soviet countries.

Country	AMC * Surveillance in NAP **	Laws and Regulations on Antimicrobial Prescription/Sales in Human Health [12,13,14,15,16]	AMC Data Used to Guide Decision-Making [12]	Reporting to International Networks [17,18,19,20,21]
**High-income countries**				
Estonia	n/a	+	+	ESAC-Net, GLASS-AMC
Latvia	+	+	+	ESAC-Net, GLASS-AMC
Lithuania	+	+	+	ESAC-Net, GLASS-AMC
Russian Federation	+	+	+	AMC network, GLASS-AMC
**Upper-middle-income countries**				
Armenia	+	+	+	AMC network, GLASS-AMC
Azerbaijan	n/a	+	+	AMC network
Belarus	n/a	+	+	AMC network, GLASS-AMC
Georgia	+	+	−	AMC network
Kazakhstan	+	+	+	AMC network
Moldova	+	+	+	AMC network, GLASS-AMC
Turkmenistan	+	+	+	n/a
Ukraine	+	+	+	AMC network, GLASS-AMC
**Lower-middle-income countries**				
Kyrgyzstan	+	+	+	AMC network
Tajikistan	+	+	+	AMC network, GLASS-AMC
Uzbekistan	n/a	+	+	AMC network

* AMC—Antimicrobial consumption surveillance; ** NAP—national action plan; n/a—data not available. Income level assigned according to World Bank [22]. “+” indicates yes; “−” indicates no.

**Table 2 antibiotics-14-00749-t002:** Antimicrobial stewardship at national level in post-Soviet countries.

Country	AMS * in NAP **	Optimising Use of Antimicrobials [12]	Integration of AWaRe Classification into NEML [12,13,15]
**High-income countries**			
Estonia	n/a	Comprehensive and sustained implementation	+
Latvia	+	Development and partial implementation	−
Lithuania	+	Development and partial implementation	−
Russian Federation	+	Comprehensive and sustained implementation	−
**Upper-middle-income countries**			
Armenia	+	Development and partial implementation	+
Azerbaijan	n/a	Development and partial implementation	−
Belarus	n/a	Comprehensive and sustained implementation	+
Georgia	+	Development and partial implementation	+
Kazakhstan	+	Comprehensive and sustained implementation	+
Moldova	+	Development and partial implementation	+
Turkmenistan	+	Development and partial implementation	−
Ukraine	+	Development and partial implementation	+
**Lower-middle-income countries**			
Kyrgyzstan	+	Development and partial implementation	−
Tajikistan	+	No implementation	−
Uzbekistan	n/a	Development and partial implementation	−

* AMS—antimicrobial stewardship; ** NAP—national action plan; n/a—unavailable data. Income level assigned according to World Bank [22]. NEML—National Essential Medicines List. “+” indicates yes; “−” indicates no.

## Data Availability

The data and links are provided in the Appendix A.

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
