# Peer review of "The Implementation of Antimicrobial Consumption Surveillance and Stewardship in Human Healthcare in Post-Soviet States: A Systematic Review"

_antibiotics, 2025, doi:10.3390/antibiotics14080749_

Round 1
Reviewer 1 Report (Previous Reviewer 1)
Comments and Suggestions for Authors
Remove the full stop at the end of your title. Line 342: You don't use Excel to generate the titles of already published works; you collate them. Moderate or minor English language editing is imperative to convey the meaning and findings of this review unambiguously.
Comments on the Quality of English LanguageModerate or minor in-house English language editing is imperative to convey the meaning and findings of this review unambiguously.
Author Response
Comment 1: Remove the full stop at the end of your title.
Response 1: Thank you for the comment. Edited as per your request.
Comment 2: Line 342: You don't use Excel to generate the titles of already published works; you collate them. Moderate or minor English language editing is imperative to convey the meaning and findings of this review unambiguously.
Response 2 : Thank you for this comment. The text was changed as follows:
[…] An Excel spreadsheet was generated for the title, abstract, and full-text screening. […]
Comment 3: Moderate or minor in-house English language editing is imperative to convey the meaning and findings of this review unambiguously.
EReviewer 3: Thank you for your comment.

Reviewer 2 Report (New Reviewer)
Comments and Suggestions for Authors
Minor Comments
- Keywords: The current keywords should be revised for alignment with Medical Subject Headings (MeSH).
- Line 46: The manuscript would benefit from referencing the One Health concept early in the Introduction to frame antimicrobial resistance within a global, interdisciplinary context.
- Table 1: The layout of Table 1 should be improved for readability. Consider redesigning it in portrait orientation, with consistent alignment and space formatting, to fit better on the page.
- Lines 124–130: Avoid describing Figure 1 in detail. Instead, highlight key insights without repeating visible data.
- Lines 136–152: The text replicates values already visible in Table 2. This section should be condensed, emphasizing only the most relevant findings.
- Lines 173–182, 186–191, 205–207: These segments clearly belong to the Discussion section rather than Results. Please relocate them to maintain structural clarity.
- Line 260: The “Semashko model” is mentioned without explanation. Add a brief definition or include a citation so the reader understands the context and relevance of this healthcare model.
- Please ensure consistent use of terms such as “antimicrobial resistance” vs. “antibiotic resistance” where applicable.
- The current Conclusions are too general. This section should be expanded to include: 1) Specific implications and reccomendations for public health; 2) Suggestions for future research or intervention models.
Author Response
Comment 1: Keywords: The current keywords should be revised for alignment with Medical Subject Headings (MeSH).
Response 1: Thank you for your comment. The keywords are changed as follows:
[…] Anti-bacterial Agents, Anti-Infective Agents, Drug Utilization, Antimicrobial Stewardship, Humans, Public health, Global health, Armenia, Azerbaijan, Belarus, Estonia, Georgia, Kazakhstan, Kyrgyzstan, Latvia, Lithuania, Moldova, Russia, Tajikistan, Turkmenistan, Ukraine, Uzbekistan […]
Comment 2: Line 46: The manuscript would benefit from referencing the One Health concept early in the Introduction to frame antimicrobial resistance within a global, interdisciplinary context.
Response 2: Thank you for the suggestion. We incorporated the text below to the paragraph you referred to introduce One Health concept:
[…] Addressing this complex challenge requires the One Health approach—a collaborative, multisectoral framework that coordinates antimicrobial resistance interventions across human, animal, and environmental health sectors.
Comment 3: Table 1: The layout of Table 1 should be improved for readability. Consider redesigning it in portrait orientation, with consistent alignment and space formatting, to fit better on the page.
Response 3: Done. Table 1 was improved according to your suggestion.
Comment 4: Lines 124–130: Avoid describing Figure 1 in detail. Instead, highlight key insights without repeating visible data.
Response 4: Done. The text was edited according to your suggestion as follows:
[…] Figure 1 reveals significant disparities in AMC surveillance implementation across the fifteen reviewed countries. The analysis demonstrates that less than half of the examined countries (40%) remain at the foundational level, suggesting substantial room for improvement in AMC surveillance capabilities. Notably, only 3 of 15 countries (20%) have progressed to comprehensive monitoring systems, underscoring that these nations can provide the most representative and consistently collected data. This uneven distribution highlights the need for targeted capacity-building initiatives to strengthen surveillance systems, particularly in countries with foundational-level implementation.
Comment 5: Lines 136–152: The text replicates values already visible in Table 2. This section should be condensed, emphasizing only the most relevant findings.
Response 5: Done. The text was edited according to your suggestion as follows:
[…] Table 2 reveals critical gaps in AMS implementation across three key domains: strategic prioritisation, antimicrobial use optimisation, and AWaRe classification adoption. While nearly three-quarters of countries have prioritised AMS strategically, other ele-ments vary significantly. Most concerning is that only four countries have achieved comprehensive antimicrobial use optimisation, with the majority remaining at partial implementation levels. Additionally, AWaRe classification adoption remains split, with slightly less than half of countries failing to integrate this WHO framework into their national essential medicines lists. These findings highlight substantial implementation gaps despite widespread strategic recognition of AMS importance.
Comment 6: Lines 173–182, 186–191, 205–207: These segments clearly belong to the Discussion section rather than Results. Please relocate them to maintain structural clarity.
Response 6: Done. Suggested segments were relocated and highlighted yellow in Discussion section.
Comment 7: Line 260: The “Semashko model” is mentioned without explanation. Add a brief definition or include a citation so the reader understands the context and relevance of this healthcare model.
Response 7: Thank you for your comment. The definition of Semashko model was added as follows:
[…] The Semashko model is a centralised, state-funded healthcare system that provides universal free medical services through a hierarchically organised network of public institutions.
Comment 8: Please ensure consistent use of terms such as “antimicrobial resistance” vs. “antibiotic resistance” where applicable.
Response 8: Thank you for your suggestion. After careful review of the manuscript several terms were edited.
Comment 9: The current Conclusions are too general. This section should be expanded to include: 1) Specific implications and reccomendations for public health; 2) Suggestions for future research or intervention models.
Response 9: Done.

Reviewer 3 Report (New Reviewer)
Comments and Suggestions for Authors
Minor corrections suggested in attached file.

Author Response
Comments:
Delete “that” – line 39.
Delete “then” – line 80.
Delete “then” – line 89.
Delete “in databases” – line 80.
Eliminate language such as “First” – line 162.
Delete “in addition” – line 230.
Delete “obvious” – line 284.
Delete “finally” – line 419.
Response: Thank you for the comments. All suggestions have been incorporated into our manuscript.

This manuscript is a resubmission of an earlier submission. The following is a list of the peer review reports and author responses from that submission.
Round 1
Reviewer 1 Report
Comments and Suggestions for Authors
The authors presented a systematic review on antimicrobial consumption surveillance and stewardship in the post-Soviet era. This should have been an interesting paper, but it has many limitations of which almost all of these are irredeemable in my view.
Below are some of my comments for the authors
Suggestions to the authors
- Title: Why not “Implementation of Antimicrobial consumption surveillance and stewardship in the former USSR countries: a systematic review”
- Lines 14-15: Delete “an” and replace “strategy” to “strategies” for proper verb agreement so the sentence reads “Antimicrobial consumption (AMC) surveillance and antimicrobial stewardship (AMS) constitute effective strategies to combat the increasing antimicrobial resistance rates worldwide”
- Line 18 and all through the manuscript: I think you should be talking of AMC surveillance, not “AMC”
- Line 23: Why human health alone? Most cases of antibiotic misuse and overuse happen in animal agriculture, particularly in the production of food-producing animals. Unfortunately, whatever goes wrong there will affect humans. I will suggest that the authors right this wrong, or at least list this as one of the limitations of this paper.
- Line 25: JBI and AACODS are undefined acronyms. Please, define these acronyms at their first use
- Lines 24-25: the word “grey” connotes uncertainty and doubtfulness in my view. Why not edit the sentence to read “Some literature from the official sites of international and national health organizations were included in the search”?
- Line 29: AMC? Comment 3 above applies
- Line 44: Please, this opening sentence. It’s complex and need not be. “Antimicrobial resistance (AMR) is a global health concern. Microorganisms….”
- Lines 82-95: I think the authors can do without these sensitive and possibly controversial claims, especially when I do not add any substantial value, in my opinion, to your paper. Kindly delete
- Lines 96-100: There’s a need for citation(s) for this categorical statement
- Lines 106-108: Please revise. “Therefore, this systematic review aims to provide current information on antimicrobial consumption surveillance and antimicrobial stewardship implementation in post-Soviet countries”.
- Line 113: Do you mean “Five hundred and thirty-four publications…” instead of “Five hundred thirty-four publications…”
- Lines 116 -117: Joanna Briggs Institute (JBI) - Comment 5 above applies
- Page 13, line 13: Korableva et al should be assigned a citation number and referenced accordingly
- Page 13, line 27: Korableva et al should be assigned a citation number and referenced accordingly
- Table 3: Should be a supplementary material in my view
- Discussion: Lines 109 -174: Are these discussion or results? The authors are supposed to be discussing implications and impacts of their findings, and not to be presenting results which should have been presented in the result section. This mix-up is a minus for this paper.
- Limitations of the research and future implications: These limitations and the additional one already suggested are too much in my view
- Section 4. 2, page 268 - Exclusion criteria: What of date of publication since the authors are talking of post USSR era? This is another costly error that further constitutes a limitation to this study.
- Lines 309 -312: how did you resolve grey areas where the two researchers couldn’t agree? The standard practice is to have a third researcher who may cast the “wining vote” in such instances. Was this done?
- I couldn’t see any information on data analysis in this systematic review. Was there any form of data analysis done by the authors?
Comments on the Quality of English Language
Moderate English language editing required
Reviewer 2 Report
Comments and Suggestions for Authors
The review paper, ‘Antimicrobial consumption surveillance and stewardship implementation in the former USSR countries: a systematic review’’, is an informative and assimilated record of the current status of Antimicrobial Usage and Antimicrobial Stewardship implementation and progress in post-Soviet countries. The study is significant as it highlights the progress, gaps and achievements on relevant aspects of GAP in this region of the world.
The paper is coherent, which makes its comprehension easy. Except for a few places, as mentioned below, it doesn’t need English editing.
I have only certain points to share.
On page 8, section 2.3 L21-22, authors have mentioned that, ‘Tajikistan reported having none or ineffective policies encouraging appropriate antimicrobial use, its availability, quality, and disposal’’, however in Table S2, it is mentioned that, (under AMS Tajikistan) Milestone: Antimicrobial stewardship programmes established in 50% of acute care facilities by Aug 2018. This means they have a stewardship program, hence doesn’t author think that their statement with regard to Tajikistan may need revision or qualification!
In Table 2. under Azerbaijan, authors write under heading Monitoring System…. antimicrobials, that No system designed to support…. use monitoring, yet under Heading Reporting AMC data…networks; it is mentioned ‘AMC’. If no system is there, then what is being reported!
Further, under Belarus and Estonia (In Table 1), Publicly available NAP is ‘No’, so AMC in AMP should be Not applicable. Also, then explain the Monitoring System for consumption…. antimicrobials for Belarus and Estonia. (This can be highlighted in text in section 2.2 Antimicrobial Consumption Surveillance for clarity).
In Section 2.3 page 8, L21-22, It is written that ‘Tajikistan reported having none or ineffective policies encouraging appropriate antimicrobial use, its availability, quality, and disposal’, although in Table S2, it is mentioned that, ‘Milestone: Antimicrobial stewardship programmes established in 50% of acute care facilities by Aug 2018’. This does not match up. Moreover, the language of first sentence also needs to be changed (cannot be written for a country!), In place of, ‘Tajikistan reported having…… use, its availability, quality, and disposal’, better, write, ‘In case of Tajikistan, no documentary evidence encouraging appropriate antimicrobial use, its availability, quality, and disposal’ was reported.
In many places in the paper, word ‘gray’ as in ‘grey literature’ appears. This word appears as grey or gray. The correct spelling is grey, so otherwise spelt word(s) needs to be replaced. For example, L111.
Abbreviations, wherever given must write full form in brackets for the first time, e.g., page 2 L47, GAP (Global Action Plan).
In results section, L108, the full test of two papers, may be the full ‘text’…
In Section 3.1.2L174, The challenges of implementing…. prioritize AMS programs, may be re-written to improve clarity of thought.
Some sentences like, on page 13, L176-180, ‘A systematic review on AMS intervention………. countries described antimicrobial surveillance as a key barrier towards developing local AMS guidelines. How?
According to the AMR surveillance conducted by Zhazykhbayeva et al. AMR surveillance in Uzbekistan and Tajikistan have unstandardized AMR surveillance, whereas Kyrgyzstan has standardized national AMR surveillance. Is this about Antimicrobial Surveillance or Stewardship. Should this be included, if yes, why?
In Fig 2, Prisma 2020 Checklist, the Box with Reports excluded is unreadable!
Reviewer 3 Report
Comments and Suggestions for Authors
This article could be interesting because it describes antibiotic stewardship in countries
that are not yet fully aligned with the rest of Europe. The problem is that it is too long
and hard to read. Lately, all researchers try to write short and fast articles. I think it would be better for the authors to divide the work
at least into two parts. Also, table 1, table 2 and table 3 can be cut or improved, everything is already reported in the Results.
I would suggest a table with all the acronyms, because given the length of the work it is difficult to remember them all
In essence, this article is hard to read and can be significantly improved, avoiding repetitions using less repetitive and heavy writing.
The quality of language is sufficiently correct.